# Hierarchical Multi-Level 3D Geometry Generation with Stress-Aware Learning

**Vadim Zlobin**                      *v@isegments.com*
*iSegments Lab*

**Vladislav Puzach**                 *vpuzach@isegments.com*
*iSegments Lab*

**Olga Bidlevich**                 *obidlevich@isegments.com*
*iSegments Lab*

**Mikhail Chetnev**                 *mchetnev@isegments.com*
*iSegments Lab*

**Vitaly Gromov**                  *vgromov@isegments.com*
*iSegments Lab*

**Reviewed on OpenReview:** *https://openreview.net/forum?id=kyoXKiyoA3*

## Abstract

Current approaches for LEGO-style 3D structural assembly are usually learned to maximize intersection over union between generated output and target construction. We propose a new approach which is able to build stable structures based on physics-aware reward. Our method employs a two-level agent architecture in which a high-level proximal policy optimization based planner proposes a scheme, while a low-level wave function collapse agent handles precise brick placement with constraint satisfaction. Experimental results demonstrate that our hierarchical method consistently constructs buildings that satisfy stress constraints while reducing material usage. We also show that replacing the finite element method solver with a Fourier neural operator achieves comparable performance, providing proof-of-concept that the proposed approach can work with neural surrogates. Our code is available at https://github.com/iSegments-Lab/stress_aware_bricks_model.

## 1 Introduction

The construction of 3D structures from discrete building elements represents a fundamental challenge in robotics, automated manufacturing, and architectural design. Traditional approaches to this problem rely heavily on topology optimization methods that require extensive computational resources and often produce solutions that are difficult to construct in practice. Recent progress, particularly with generative models and reinforcement learning, has enabled the creation of complex shapes and assemblies (Chung et al., 2021; Kolodiazhnyi et al., 2025). However, a significant gap remains between generating objects that are visually plausible and those that are physically viable. Most current approaches to LEGO brick construction focus on replicating a target shape, commonly employing geometric metrics such as intersection over union (IoU) as the main optimization criterion (Chung et al., 2021). Although effective within their intended scope, these methods do not ensure that the resulting structure can support its own weight or resist external forces. We focus on the complex and application-driven challenge: structurally-informed combinatorial construction. The objective moves past simply assembling bricks to match a shape, aiming instead to create a stable configuration that complies with physical principles, reduces material usage, and maintains structural integrity under load. This problem introduces a new layer of complexity, as the validity of an action depends not only

on local geometric constraints (e.g., element non-intersection) but also on its global impact on the entire structure's stress distribution.

The key insight driving our work is that effective structural construction requires reasoning at multiple levels of abstraction: strategic planning for overall structural layout and detailed execution for precise element placement. This naturally suggests a hierarchical approach in which different agents operate on different scales and optimize for different objectives. Our high-level planner focuses on strategic decisions about load distribution and structural stability, while our low-level executor ensures that individual brick placements satisfy local constraints.

Major Contributions:

- Novel Problem Formulation: We introduce stress minimization as a primary objective for 3D construction, moving beyond shape reconstruction to consider structural integrity and resource efficiency.

- Hierarchical Agent Architecture: We present a two-level system that integrates Proximal Policy Optimization (PPO) based strategic planning with Wave Function Collapse (WFC) based constraint satisfaction, enabling both global optimization and local constraint enforcement.

- Physics-Integrated Rewards: Our approach incorporates realistic stress analysis using finite element method (FEM), providing physics-based feedback for structural optimization.

- Efficient Constraint Handling: Using WFC for low-level execution, we achieve efficient constraint satisfaction without the computational overhead of traditional planning approaches.

We validate our framework through a series of experiments that test its core components. Our experiments demonstrate that the hierarchical 3D planner achieves higher success rates compared to a simplified 2D version, confirming that full spatial reasoning is critical to achieving structural stability. We then show that the 3D planner's high performance is maintained when a fast neural operator replaces the traditional physics solver, establishing the viability of our approach for scalable, structurally-aware construction.

## 2 Related Work

Recent advances in reinforcement learning (RL) based construction have focused primarily on LEGO brick assembly tasks. Brick-by-Brick introduced a pioneering approach that uses graph neural networks and action validity prediction for sequential brick placement, optimizing for IoU with target shapes (Chung et al., 2021). Their work demonstrated that incomplete target information (2D images) could be sufficient for constructing complex 3D objects through a comprehensive understanding of partial information and long-term planning, but was limited to shape reconstruction objectives. Budget-aware construction was explored in BrECS, which combines U-shaped convolutional networks with efficient constraint satisfaction for sequential brick assembly (Ahn et al., 2024). Although this approach showed improvements in assembly speed and constraint handling, it remained focused on shape completion rather than structural optimization. More recently, multimodal approaches have emerged with CADrille, which combines point clouds, images, and text for computer-aided design reconstruction (Kolodiazhnyi et al., 2025). However, these approaches primarily target design reconstruction rather than structural construction with physical constraints. A framework Kakooee & Dillenburger (2024) leverages PPO algorithm Schulman et al. (2017) to optimize space layouts, showing how RL can accommodate offline tasks and seamlessly integrate with existing computer-aided design software.

In addition to learning-based construction methods, the WFC algorithm has emerged as a powerful tool for procedural content generation, particularly in scenarios that require the adherence to local constraint rules. Originally developed for tile-based image generation, WFC operates by maintaining a superposition of possible states for each cell in a grid and progressively collapsing these possibilities based on local compatibility constraints (Gumin, 2016). The effectiveness of the algorithm arises from its ability to generate coherent global patterns from simple local rules. Newgas (2020) provides a comprehensive explanation of WFC as a constraint programming approach, where the computer uses built-in algorithms to find solutions to rigorously

defined problems rather than following explicit imperative instructions. Recent developments in WFC have extended its applicability to 3D construction scenarios and multiscale generation problems. The integration of WFC with other procedural generation techniques has shown particular promise in creating complex, rule-constrained structures while maintaining computational efficiency.

While construction planning determines placement strategies, accurate structural evaluation requires physics-based models. The integration of physics-based constraints into neural network architectures has revolutionized computational mechanics and structural analysis. Physics-Informed Neural Networks (PINNs) have demonstrated remarkable success in solving partial differential equations while incorporating domain knowledge directly into the learning process (Zhang et al., 2024; Bolandi et al., 2023; Kim et al., 2025). Fourier Neural Operators (FNOs) represent a significant advancement in this field, enabling the learning of operators that map between infinite-dimensional function spaces (Li et al., 2021; Kovachki et al., 2023). The application of FNOs to the prediction of the stress-strain field in composite materials has shown exceptional promise (Rashid et al., 2022; Khorrami et al., 2024; Shin et al., 2025).

## 3 Preliminary

**Action Masking**  Action masking in RL is a mechanism that sets the probability of invalid action to zero before sampling, so the policy can select only meaningful actions. This removes the need for constraint-violation penalties in the reward design. In our work, action masking enforces all constraints (support, non-overlap, boundary, structural integrity) at each step.

**WFC**  A constraint-solving algorithm for procedural grid generation. Each iteration consists of three steps: *observe*, *collapse*, and *propagate*. The observe step selects which uncollapsed cell to process next. The collapse step samples a tile from that cell's admissible set, where each tile has its own weight during sampling. The propagate step updates the admissible set of each neighbor, removing tiles that are incompatible with the collapsed cell. A *contradiction* occurs when propagation eliminates all tiles from some cell's admissible set, making any valid completion of the grid impossible.

**FEM and FNO**  FEM is a numerical method that discretizes the structure into finite elements to approximate the solution of the PDEs. The *von Mises stress* $\sigma_{vm}$ is a scalar engineering criterion that aggregates the full 3D stress tensor into a single non-negative number. Both FEM and FNO produce a stress field $\sigma_{vm}$ for a given structure $s$, but differ algorithmically. FEM discretizes $s$ into a mesh and, per instance, assembles and solves a sparse linear system

$$\mathbf{K}(s)\,\mathbf{u} = \mathbf{f}(s), \qquad \boldsymbol{\sigma} = \mathbb{C} : \tfrac{1}{2}\big(\nabla\mathbf{u} + (\nabla\mathbf{u})^\top\big),$$

where $\mathbf{K}(s)$ is the stiffness matrix and $\mathbf{f}(s)$ encodes body and surface loads; the von Mises stress is then recovered pointwise as

$$\sigma_{vm} = \sqrt{\tfrac{1}{2}\big((\sigma_1 - \sigma_2)^2 + (\sigma_2 - \sigma_3)^2 + (\sigma_3 - \sigma_1)^2\big)},$$

with $\sigma_1, \sigma_2, \sigma_3$ the principal stresses of $\boldsymbol{\sigma}$. FNO instead learns a single operator $\mathcal{G}_\theta$ offline and predicts the field in one forward pass:

$$\sigma_{vm} \approx \mathcal{G}_\theta\big(\mathrm{SDF}(s)\big), \qquad \mathcal{G}_\theta = Q \circ \mathcal{L}_L \circ \cdots \circ \mathcal{L}_1 \circ P,$$

$$\mathcal{L}_\ell(v_\ell) = \phi\big(Wv_\ell + \mathcal{F}^{-1}\big(R_\ell \cdot \mathcal{F}(v_\ell)\big)\big),$$

where $P$ lifts the scalar SDF input to a hidden-width feature field, $Q$ projects the final hidden field to the scalar stress output, $\mathcal{F}$ is the discrete Fourier transform, $R_\ell$ are learnable spectral weights truncated to the lowest $k$ Fourier modes, $W$ is a pointwise linear map, and $\phi$ is a nonlinearity. The trade-off is between solving a PDE from scratch for each input (FEM) and learning an operator once and then evaluating it at inference time (FNO).

# 4 Problem Formulation

## 4.1 Construction Environment

We define our construction environment as a bounded 3D space with coordinates $(x, y, z)$ of size $L \times W \times H$ discretized into unit cells, as illustrated in Figure 1. The construction process involves the sequential placement of LEGO-style bricks with fixed dimensions $(l_b, w_b, h_b)$ in our base configuration. We use the term "LEGO-style" to describe the geometric and connectivity paradigm: discrete modular units arranged on a stud grid. The actual domain is abstract modular structural assembly.

Each placement must satisfy fundamental constraints:

1. **Support constraint**: bricks at height $z > 1$ must be supported by at least one existing brick.

2. **Non-overlap constraint**: new bricks cannot occupy space already filled by existing bricks.

3. **Boundary constraint**: all bricks must lie entirely within the construction space.

4. **Structural integrity**: ensure that the structure forms a connected component.

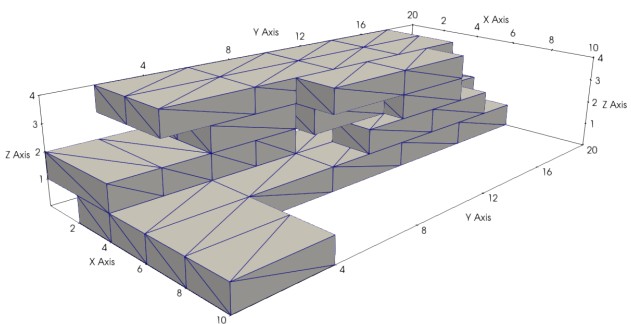

Figure 1: A valid construction. The configuration of $4 \times 2 \times 1$ bricks satisfies all support, non-overlap, boundary, and connectivity constraints.

The scene is represented as a binary occupancy grid of size $L \times W \times H$, where each cell contains a value of 1 if it is occupied by a brick and 0 otherwise. Due to the hierarchical nature of our approach, the state and action spaces for the high- and low-level agents are distinct and are described in detail in Sections 5.3 and 5.4, respectively.

## 4.2 Optimization Objective

Unlike previous work Chung et al. (2021); Ahn et al. (2024) that optimizes for shape similarity, we formulate construction as a multiobjective optimization problem. Let $s$ denote a candidate structure, and let $\Omega$ be the set of structures satisfying the constraints defined earlier in Section 4.1. We define three scalar objective functions:

$$f_1(s) = \sigma(s) = \max_{\text{brick} \in s} \sigma_{vm}(s, \text{brick}),$$
$$f_2(s) = |h(s) - h_{\text{target}}|,$$
$$f_3(s) = N_{\text{brick}}(s),$$

where $h(s)$ is the height of the structure, $h_{target}$ is the target height, $N_{\text{brick}}(s)$ is the number of placed bricks, and $\sigma_{vm}(s, \text{brick})$ is the von Mises stress of a brick in structure $s$ – a single number that measures how large the mechanical stress is in that brick; lower values indicate a safer structure. We describe the computation of this stress using FEM in Section 5.2.

The feasible set is first restricted by a material stress threshold $\sigma_{\text{thld}}$ and target height:

$$\Omega^* = \left\{ s \in \Omega \mid f_1(s) < \sigma_{\text{thld}}, \ f_2(s) = 0 \right\} \tag{1}$$

and among all structures in $\Omega^*$, we search for one that minimizes material usage:

$$s^* = \arg \min_{s \in \Omega^*} N_{\text{brick}}(s) \tag{2}$$

## 5  Proposed Method

Our training is a two-stage cycle of data collection and policy updates, where each episode produces a complete candidate structure. An overview of the full pipeline is shown in Figure 2, and the episode structure is given in Algorithm 1. We consider two planner modes, that differ in planning horizon and feedback frequency.

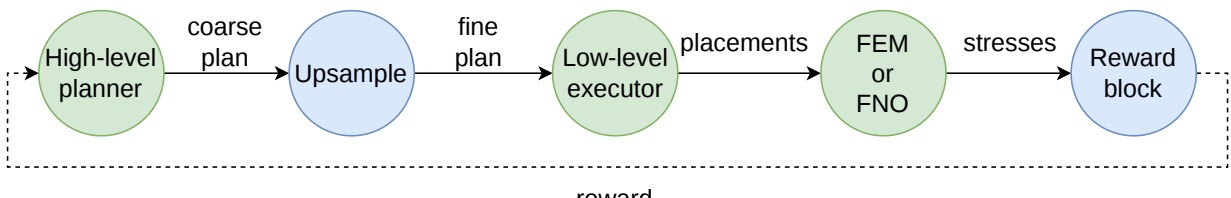

Figure 2: Overview of the proposed pipeline.

The 2D approach operates layer-by-layer: the high-level policy proposes a plan, the WFC executor builds it, the policy observes the result, and plans the next layer. Since each layer is a foundation for the next, bottom-up construction makes the feedback semantically meaningful. Moreover, frequent feedback should help the 2D planner compensate for executor errors.

In contrast, the 3D method is open-loop: the entire structure is planned at once and then executed. This is motivated by the cross-layer nature of mechanical stress: a brick placed near the top of the structure influences the load paths several layers below.

### 5.1  Reward Design

We compute the reward after the complete structure is built (episode end) by running the stress evaluation on the final assembly; intermediate steps receive zero reward. Structural integrity is enforced by the use of action masking. The reward function encodes priorities from (1)–(2):

$$R_\sigma = \begin{cases} 1, & \text{if } \big(\sigma(s) < \sigma_{\text{thld}}\big) \wedge \big(h = h_{\text{target}}\big) \\ 0, & h < h_{target} \\ 0.5 \cdot \exp\left(\dfrac{\sigma_{\text{thld}} - \sigma(s)}{\eta \cdot \sigma_{\text{thld}}}\right) & \text{otherwise} \end{cases} \qquad \text{(Stress reward)}$$

$$R_h = w_h \cdot \max\left(0, \frac{h_{\text{target}} - |h_{\text{target}} - h|}{h_{\text{target}}}\right) \qquad \text{(Height reward)}$$

$$R_{\text{vol}} = \begin{cases} w_{vol} \cdot \dfrac{V_{\max} - N_{\text{brick}}}{V_{\max} - h_{\text{target}}}, & \text{if } \big(\sigma(s) < \sigma_{\text{thld}}\big) \wedge \big(h = h_{\text{target}}\big) \\ 0, & \text{otherwise} \end{cases} \qquad \text{(Volume reward)}$$

$$R = \begin{cases} 1 + R_{\text{vol}}, & \text{if } \big(\sigma(s) < \sigma_{\text{thld}}\big) \wedge \big(h = h_{\text{target}}\big) \\ R_\sigma + R_h, & \text{otherwise} \end{cases} \tag{3}$$

where $\eta \in [1, \infty)$ controls the sharpness of the reward functions for structures that exceed the stress threshold, $w_{vol} \in \mathbb{R}^+$ is a weight for volume-based reward component, $w_h \in (0, 0.5]$ balances the height objective against other, $V_{max}$ is the maximum allowed number of bricks.

The design of the reward function is essential to guide the agent toward desirable solutions. Our parameter choices are guided by a two-stage learning strategy: first, achieve structural viability, and second, optimize for resource efficiency.

**Structural Viability**  The stress threshold $\sigma_{\text{thld}}$ defines the boundary for a structurally sound design. This threshold is derived from the properties of the brick material and any structure whose maximum von Mises stress exceeds this value is considered failure. The height and stress rewards ($R_h$, $R_\sigma$) reflect the primary goal of training an agent that can successfully build a structure of the target height while meeting the stress constraint. The reward for any "unsuccessful" state (either $h < h_{target}$ or $\sigma >= \sigma_{thld}$) is structured to be strictly less than the reward for a "successful" state. A successful construction receives a base reward of 1, plus a volume bonus. We set $w_h \leq 0.5$ to ensure that even if an agent builds to the full target height (earning the maximum $R_h = w_h$), but fails the stress test (earning $R_\sigma < 0.5$), the total reward ($R = R_\sigma + R_h = R_\sigma + w_h$) remains definitively below the success reward of 1.0. This gradient strongly incentivizes the agent to first satisfy the hard constraints.

**Resource Efficiency**  The volume reward $R_{vol}$ acts as a bonus reward, applied only after a successful construction is achieved. It introduces the trade-off between resource economy and structural robustness, encouraging the agent to use the fewest bricks possible while potentially creating solutions that are closer to the stress threshold.

## 5.2  Physics Integration

To evaluate structural stability, we develop a finite element model that simulates realistic loading scenarios for brick assemblies. The model incorporates two primary mechanisms: gravitational body forces applied throughout the volume and a uniform surface traction $\tau$ applied to the upper surface to simulate operational loads (e.g., payload). As illustrated in Figure 3, the boundary conditions are defined by fixed displacement constraints at the base ($\mathbf{u} = \mathbf{0}$) to represent foundation support, while all lateral boundaries remain free.

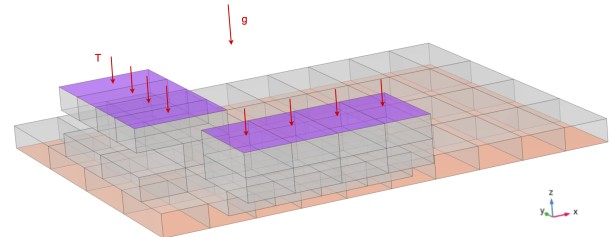

Figure 3: Boundary conditions.

The structural analysis is governed by the linear elasticity equations under static equilibrium conditions:

$$\nabla \cdot \boldsymbol{\sigma} + \rho \mathbf{g} = 0$$
$$\boldsymbol{\sigma} = \mathbb{C} : \boldsymbol{\epsilon}$$
$$\boldsymbol{\epsilon} = \frac{1}{2} \left( \nabla \mathbf{u} + (\nabla \mathbf{u})^T \right)$$

The von Mises stress is given by:

$$\sigma_{vm} = \sqrt{\frac{(\sigma_1 - \sigma_2)^2 + (\sigma_2 - \sigma_3)^2 + (\sigma_3 - \sigma_1)^2}{2}}$$

where $\boldsymbol{\sigma}$ represents the stress tensor, $\rho$ is material density, $\mathbf{g}$ is the gravitational acceleration vector, $\mathbb{C}$ is the fourth-order elasticity tensor, $\boldsymbol{\epsilon}$ denotes the strain tensor, and $\mathbf{u}$ is the displacement field. The constitutive relationship assumes isotropic linear elastic behavior with material properties.

The physical validity of structures is verified using a finite element solver. Stress analyses were performed asynchronously through a custom cloud service employing DOLFINx (FEniCS project). This asynchronous approach enabled parallel processing of multiple structural configurations, significantly reducing computational overhead during reinforcement learning training. For additional validation and benchmarking, selected configurations were also analyzed using COMSOL Multiphysics to verify consistency of predictions across different solvers. Mesh resolution was chosen to balance computational efficiency with accuracy, prioritizing rapid feedback and scalability over high-fidelity resolution.

However, relying solely on traditional FEM solvers for the reward calculation in a reinforcement learning loop can introduce computational bottlenecks. To explore potential solutions, we implement a physics approximator based on FNOs, which provides rapid and accurate stress field predictions from geometric data.

Our physics approximator leverages a 3D FNO architecture tailored to predict stress fields from signed distance function (SDF) representations of the built structures. An SDF assigns to each voxel a scalar equal to its signed distance to the nearest surface. The model processes a single-channel 3D SDF input through a multi-layer FNO core with a fixed number of Fourier modes and hidden channels, followed by a projection head that outputs a single-channel von Mises stress field over the domain.

The trained FNO model replaces the traditional FEM solver in the RL environment's reward pipeline. First, the current assembly state is converted into its SDF representation. The FNO then predicts the full 3D von-Mises stress field, after which stress metrics are extracted from the prediction to calculate the reward. This approach enables the integration of realistic physics into RL for structural assembly without compromising fidelity. It demonstrates that the RL agent can learn to build minimal-stress 3D structures from bricks using a neural surrogate.

### 5.3 High-Level Planner

Operates on a coarse binary occupancy grid $\tilde{s} \in \{0,1\}^{\tilde{L} \times \tilde{W} \times H}$. For the 2D planner, the observation concatenates the downsampled occupancy of the full scene and the current coarse plan for the next layer, which is updated after each action. For the 3D planner, the observation is the coarse plan array alone. In our experiments, $(L, W, H) = (20, 20, 4)$ and $(\tilde{L}, \tilde{W}, H) = (10, 10, 4)$.

The planner produces a coarse plan $\tilde{P} \in \{0,1\}^{\tilde{L} \times \tilde{W} \times H}$ that marks which coarse cells should be filled with bricks. Then, this plan is upsampled to fine resolution $(L, W, H)$. The relationship between these resolutions is defined by the integer scaling factors $k_L = \frac{L}{\tilde{L}}$ and $k_W = \frac{W}{\tilde{W}}$. For any cell on the fine grid, the value is determined by the corresponding coarse cell:

$$P_{x,y,z} = \tilde{P}_{\lfloor x/k_L \rfloor, \lfloor y/k_W \rfloor, z}$$

$P_{x,y,z}$ guides the low-level WFC executor, which operates only on the fine grid. We intentionally define the planner's action space on the coarse grid, so that each action corresponds to a large shape that can cover multiple bricks; this biases the high-level policy towards strategic decisions rather than brick-level placement.

We use the actor–critic architecture, which is illustrated in Figure 4. At each step, the policy selects a staged action $a = (a_{color}, a_{shape}, a_{center})$. A shared 3D convolutional encoder processes the 3D scene, and the resulting features are passed through an MLP to produce actor features. Using actor features, the policy first samples the block type $a_{color} \in \{0,1\}$; if $a_{color} = 1$, the selected shape region is marked as a usable space for bricks; otherwise, the region is treated as empty (no bricks placed). The policy then samples a shape index $a_{shape}$ from a masked categorical distribution; the shape logits are computed from a concatenation of the actor features and a learned projection of the color logits. Finally, the policy samples the placement center from a distribution whose logits are computed from a concatenation of the actor features, the color embedding, and the shape logits. See Appendix B (Tables 4 and 5) for full architectures.

$a_{shape}$ is an index in a predefined shape library. Each shape in this library satisfies three criteria: it must be rectangular, it must fit within a coarse grid, and it is made up of several bricks. Thus, the library contains

---

**Algorithm 1** Training episode structure

---

**Require:** Max levels $H$, max steps $N$, WFC restarts $K$
**Ensure:** Replay buffer $\mathcal{B}$
1:   $M \leftarrow \mathbf{1}^{L \times W}$       ▷ Support mask; all cells grounded at level 1
2:   **for** level $= 1$ **to** $H$ **do**       ▷ Outer loop omitted for 3D planner
3:      $s \leftarrow s_0$;   done $\leftarrow$ False;   step $\leftarrow 0$
4:      **while** not done **and** step $< N$ **do**
5:         $a = (a_{\text{color}}, a_{\text{shape}}, a_{\text{center}}) \sim \pi_\theta(s)$       ▷ Masked categorical
6:         $s'$, done $\leftarrow Env.Step(s, a)$       ▷ Updates coarse plan $\tilde{P}$
7:         $\mathcal{B}.Add(s, a, r{=}0, s', \text{done})$;   $s \leftarrow s'$;   step $+= 1$
8:      $P_{x,y} \leftarrow \tilde{P}_{\lfloor x/k_L \rfloor, \lfloor y/k_W \rfloor}$       ▷ Upsample coarse plan to fine grid ($L \times W$)
9:      placements $\leftarrow WFC(P, M, K)$       ▷ Algorithm 2
10:     $Env.Apply(placements)$; $Env.ComputeStress()$
11:     $\mathcal{B}.AssignReward(Env.ComputeReward())$       ▷ Episode-level reward to last transition
12:     $M \leftarrow Env.ExtractSupport(\text{level})$       ▷ Support mask for next level

---

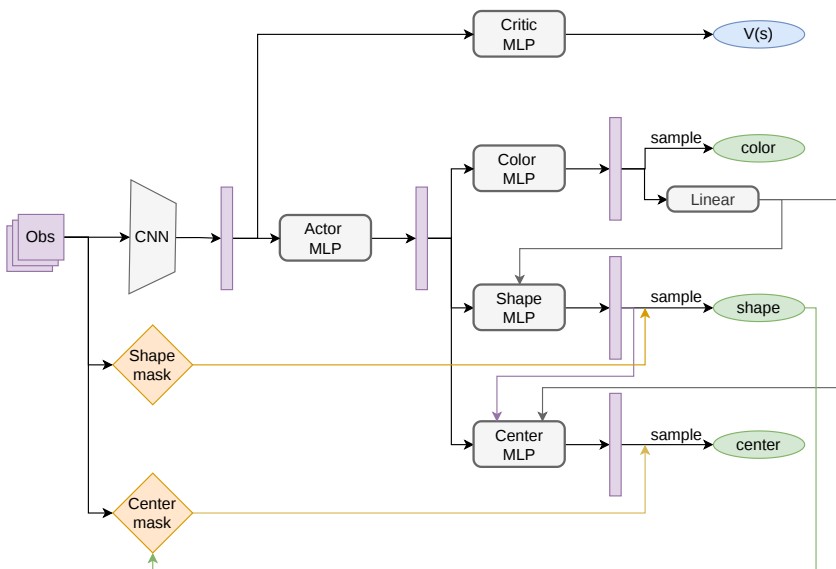

Figure 4: Actor–critic architecture for the high-level planner; the multi-head actor produces $a_{color}$, $a_{shape}$, and $a_{center}$ outputs with sequential conditioning.

the base primitives (e.g. $4 \times 2$, $2 \times 4$), two-brick composites (e.g. $2 \times 8$, $4 \times 4$, $2 \times 8$), three-brick composites (e.g. $4 \times 6$, $6 \times 4$) and other larger structures, provided that they do not exceed the boundaries. For the 3D planner, each of these 2D shapes is extruded along the vertical axis (e.g., the shape $4 \times 2$ expands to $4 \times 2 \times 1, \ldots, 4 \times 2 \times H$).

Feasibility constraints are enforced via action masking: for components $a_{shape}$ and $a_{center}$, a boolean mask is computed from the current scene state (for $a_{center}$: also from sampled $a_{shape}$) and applied by a masked categorical distribution. The center action is not selected from the full grid $\{1, \ldots, L\} \times \{1, \ldots, W\}$, instead it selects from a set of valid candidate centers, so the boundary coordinates (e.g. (1,1)) may be invalid because the shape would exceed the scene.

An episode terminates when no feasible placement remains: given the current occupancy grid and the predefined shape library, the action mask becomes empty (i.e., no shape fits in any free region without violating constraints).

### 5.4 Low-Level WFC Executor

The low-level execution is performed using a modified WFC algorithm specialized for brick assembly. It operates on a fine grid divided into frames of fixed size $F \times F$. The primary responsibility of the WFC executor is to handle local constraint satisfaction while maximizing the Intersection over Union (IoU) with the high-level strategic plan. We treat each brick as a composite object consisting of $1 \times 1$ tiles. In total, we use 17 tile types: code 0 denotes empty space, codes 1–8 correspond to a vertically oriented brick, and codes 9–16 correspond to a horizontally oriented brick.

Algorithm 2 consists of two stages: grid initialization, in which each cell is assigned a set of admissible tiles and their weights, and an iterative execution loop that adjusts tile weights, collapses a cell, and propagates the resulting constraints. Cells are processed in a predetermined sequential order (left to right, top to bottom) rather than by an entropy-based heuristic, since this ordering prevents the generation of disconnected components.

We use two types of rules. The tile-level rules for a cell $c = (x, y, z)$ examine its neighbors $(x + 1, y, z)$, $(x - 1, y, z)$, $(x, y + 1, z)$, $(x, y - 1, z)$, and $(x, y, z - 1)$, and impose hard constraints by removing incompatible tiles from the admissible set (lines 2, 6 and 23 in Algorithm 2). Brick-level rules evaluate constraints at the level of entire bricks rather than individual tiles. For a given brick $b$, we define the following rules: $Consistent(b), Solid(b), Based(b)$. $Solid(b)$ checks that the brick fits within the boundaries of the scene. $Consistent(b)$ verifies the compatibility with already collapsed neighbors and the high-level plan $P$, applying a penalty to tiles that extend beyond the plan. The $Based(b)$ rule requires that at least one brick tile is supported. This rule ensures that WFC will only build structures for which the support constraint is satisfied. A more detailed description of the tile-level and brick-level rules, as well as Figure 12 with the tile encoding scheme, are provided in the Appendix C.

## 6 Experiments

The implementation utilizes JAX/Flax for efficient GPU acceleration and automatic differentiation.

### 6.1 Experimental Setup

We train the high-level planner with PPO; the full set of PPO hyperparameters is given in Appendix B / Table 3. The high-level agent operates on a $10 \times 10 \times 4$ grid; the low-level agent acts on a $20 \times 20 \times 4$ grid with WFC frame size $F = 10$. The target height is 4 units, episodes last up to 25 steps, and training uses 20 parallel environments with reward weights $w_{vol} = w_h = 0.5$. The shape library consists of 16 brick types: 4×2, 2×4, 8×2, 4×4, 2×8, 6×4, 4×6, 8×4, 4×8, 10×4, 4×10, 8×6, 6×8, 8×8, 10×8, 8×10.

Physics simulation uses a Dolfinx-based solver with aluminum (Young's modulus $E = 68$ GPa, Poisson's ratio $\nu = 0.30$, density $\rho = 2698.9$ kg/m$^3$), a uniform applied force of $3 \times 10^6$ N, stress threshold $\sigma_{thld} = 4 \times 10^8$ Pa, and mesh factor 0.7. We set $\sigma_{thld} = 4 \times 10^8$ Pa (400 MPa) as our baseline, corresponding to a conservative yield strength for high-strength aluminum alloys. To demonstrate the physical range of the problem, lower-strength structural steel typically has a yield of 250 MPa, while ABS plastic has a yield strength of approximately 40–45 MPa. The parameters of the aluminum material are used to define a physically grounded stress benchmark rather than to target a specific material.

We compare our method to a simplified baseline based on a 2D high-level planner. This baseline restricts high-level decisions to two spatial dimensions: at each height layer, it plans the placement of components without reasoning about full 3D connectivity between layers. The rest of the pipeline—including low-level brick placement using WFC, constraint enforcement, and physics-based reward functions—remains identical. This controlled comparison directly isolates the benefits of full spatial (3D) reasoning for structural stability, efficient material usage, and stress minimization.

---

**Algorithm 2** WFC Brick Assembly Executor

---

**Require:** Plan P, support mask M (from layer below, first level: 1 for all cells)
**Ensure:** Set of brick placements (x, y, level, orientation)
 1: **for** each cell $c = (i, j)$ of grid $G$ **do**                                    ▷ Initialize grid $G$ of size $F \times F$
 2:     $c.T \leftarrow$ tiles compatible with scene borders                           ▷ Tile-level rules, hard constraint
 3:     **if** $P(i, j) == 0$ **then**
 4:         $w(\text{space}) \leftarrow 0.9999$, $w(\text{other}) \leftarrow 0.0001/(|c.T| - 1)$
 5:     **else**
 6:         $w(\text{space}) \leftarrow 0.0001$, tiles incompatible with zero neighbors of $c$ get lower weight         ▷ Tile-level
 7: **for** attempt $= 1$ **to** $K$ **do**
 8:     Reset $G$ to fresh initialized grid                                          ▷ After contradiction
 9:     **for** each cell $c = (i, j)$ in specified order **do**
10:         **if** $P(i, j) == 0$ **then**
11:             Keep current weights
12:         **else**
13:             potential $\leftarrow 0$
14:             **for** tile $t$ with $t.\text{code} \neq 0$ **do**
15:                 b $\leftarrow$ brick induced by tile
16:                 coeff $\leftarrow \min(1, \text{Consistent}(b), \text{Solid}(b), \text{Based}(b))$           ▷ Brick-level rules
17:                 $w(t) \leftarrow w(t) \times \text{coeff}$, potential $\leftarrow$ potential + coeff           ▷ Soft constraint
18:             **if** potential $< 0.2$ **then**
19:                 $w(\text{space}) \leftarrow 0.8$
20:         **if** $c.T$ is empty **then**
21:             **break**                                                            ▷ Contradiction $\rightarrow$ outer loop
22:         Collapse $c$: $t = \text{argmax } w$                                      ▷ greedy collapse
23:         Propagate: filter each neighbor's tiles to those compatible with $c$      ▷ Tile-level
24:         **if** any neighbor has empty tile set **then**
25:             **break**                                                            ▷ Contradiction
26:     **if** grid $G$ successfully collapsed **then**
27:         **return** placements extracted from $G$                                 ▷ Success: break after first valid

---

## 6.2 Neural Operator Training

When evaluated as a separate component on identical scenes, the FNO approximator achieves a $85\times$ to $150\times$ speedup over conventional FEM. However, when measuring the end-to-end iteration time, FEM and FNO yield comparable results, since the training pipeline was originally optimized for CPU-based FEM, and the physics computation comprises only 18% of the per-iteration time.

Training data consists of paired SDF and FEM-derived stress maps from a diverse set of brick assembly configurations (20–200 bricks in various spatial layouts). The Assemblies are labeled "good" if stress is below $4 \cdot 10^8$ Pa and "bad" otherwise; stratified sampling ensures a balanced representation across brick counts and stress categories. Stress values are logarithmically scaled to stabilize training.

The FNO approximator is configured with 16 Fourier modes per spatial dimension to capture the relevant stress gradients across the $64 \times 64 \times 12$ discretization. All architectural and optimization hyperparameters are listed in Appendix B / Table 6.

Qualitatively, the predicted and true values are very similar: both exhibit negligible stress values in the first and second stages, with maximum stresses concentrated on the upper floor (see Figures 5 and 6). The high-stress clusters (red/hot regions) match well in location and shape, though predicted stress decreases more smoothly compared to sharper decay in true data. Quantitatively, the values are also close: the maximum predicted stress is $8.2 \times 10^8$, while the maximum true stress is $8.9 \times 10^8$. This corresponds to a relative error

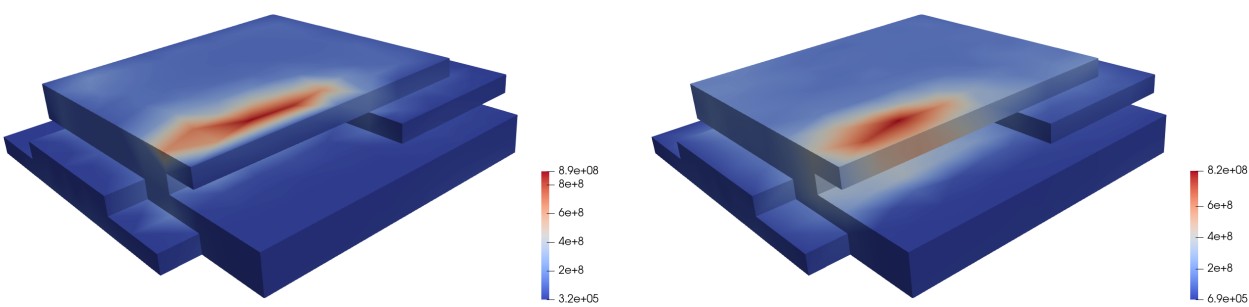

Figure 5: Von Mises stress values computed by the FEM solver

Figure 6: Predicted von Mises stress values by the neural operator

of 7.9% in predicting maximum stress, which is acceptable for the calculation of rewards in the reinforcement learning algorithm.

## 6.3 Structural Performance Results

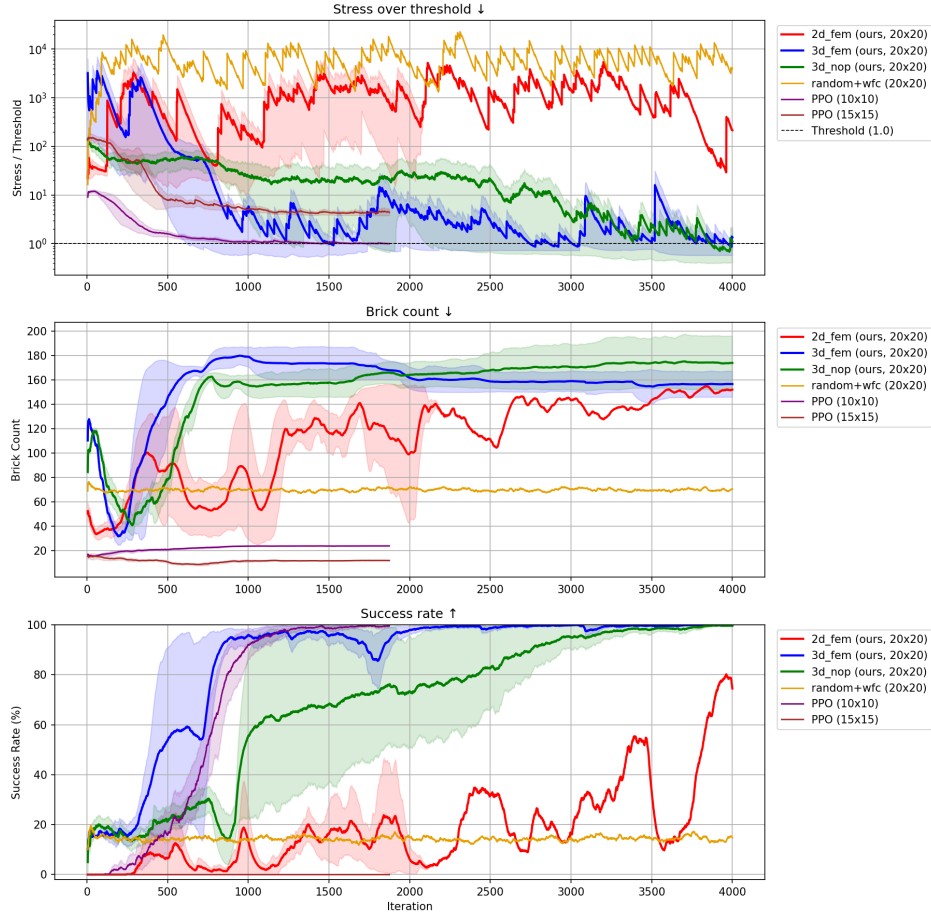

Figure 7: Training performance of the proposed hierarchical methods (2D+FEM, 3D+FEM, 3D+FNO) on the $20 \times 20$ task versus a Random+WFC baseline and non-hierarchical PPO baseline trained on smaller scenes ($10 \times 10$, $15 \times 15$).

To evaluate the effectiveness of our method, we compare it against two distinct baselines. The first is a naive approach: the high-level planner uses a random strategy with action masking, while the low-level execution is still handled by WFC. Second, to isolate the contribution of hierarchy itself, we introduce a non-hierarchical baseline trained with PPO. In this setting, the policy directly predicts a brick placement $(x, y, z, \text{rotation})$ at each step, where $x, y$ are the coordinates of the brick's center, $z$ is the layer index, and $\text{rotation} \in \{0°, 90°\}$. We evaluate this flat PPO on smaller $10 \times 10$ and $15 \times 15$ scenes to show its scaling behavior.

Figure 7 presents the main quantitative results of our experiments, comparing the performance of our 2D planner, 3D planners (with the FEM and the FNO approximator) against the random baseline in three key metrics: structural stability, resource efficiency, and overall construction success. For the learned agents, solid lines represent the mean performance averaged in three independent runs with different random seeds, while shaded areas show the corresponding minimum and maximum values of those runs, illustrating the performance variance. Each plotted point is computed on episodes freshly sampled from 20 parallel environments per seed; the curves therefore reflect performance on unseen rollouts. Because the environment specification (grid size, target height, shape library, material parameters) is fixed, there is no separate held-out distribution. The random baseline was evaluated in a single run, as it shows nearly identical stable performance in 4,000 iterations.

**Structural stability (stress over threshold).** The primary indicator of a successful design is the stress-to-threshold ratio, shown in the top plot. A value below 1.0 signifies that the maximum von Mises stress is within the acceptable material limits. As illustrated, the naive approach exhibits the worst performance, fluctuating around a value of 10,000. The 2D planner only manages to drop below 100 toward the end of training. Flat PPO successfully solves the $10 \times 10$ task, consistently reaching a stress value below the threshold. However, on the larger $15 \times 15$ task, the algorithm gets stuck in a local optimum where stress values remain above the threshold because it converges to solutions with too few bricks. By contrast, both hierarchical 3D planners reduce the stress ratio below 1.0 on the $20 \times 20$ task. A closer comparison reveals that the agent guided by the FEM solver exhibits faster convergence, achieving acceptable performance in fewer than 3,000 PPO iterations. Nevertheless, the FNO-based agent also reliably converges to the required performance level.

**Overall performance (success rate).** The bottom plot provides the percentage of episodes in a batch that result in a complete structure (that is, the target height is met and the stress-to-threshold ratio is below 1.0). This metric offers a decisive look at overall performance. Both 3D planners (blue and green) converge to near-perfect success rates, with the operator-based agent requiring more interactions to do so. In contrast, the 2D planner exhibits strong fluctuations and peaks at only about 80%, underscoring its instability and limited effectiveness. The flat PPO performs differently depending on scale: achieves near perfect result on $10 \times 10$, but converges to 0% in $15 \times 15$ case, showing that the non-hierarchical approach does not scale beyond small scenes. The naive baseline remains nearly constant at a success rate of roughly 17%, showing that perfect performance requires learned structural heuristics rather than relying on valid random placements.

**Resource efficiency.** The middle plot shows the number of bricks used by each agent. This metric reveals the trade-off between the resource economy and structural integrity. Although using fewer bricks is efficient, building a stable, full-height structure is difficult with too few materials. In contrast, a high brick count does not guarantee low stress. The Random+WFC baseline maintains an almost flat brick count of roughly 70 bricks throughout the training. In contrast, the 2D planner demonstrates an upward trend in training, which is consistent with its low success rate: because unsuccessful episodes do not receive the volume bonus, there is little incentive to reduce brick usage, while the stress-driven component still rewards reductions in peak stress—often achieved by adding bricks. By contrast, the 3D planner's brick count begins to decline after roughly 1,000 iterations, coinciding with near-perfect success rates; this situation indicates that the volume bonus starts to dominate the optimization, guiding the agent to reduce material while maintaining structural feasibility.

These results highlight the superiority of the hierarchical 3D planning approach over both baselines. A flat PPO policy is sufficient in the small $10 \times 10$ setting, but on the $15 \times 15$ scene it converges to a local optimum without ever crossing the threshold. This also provides strong evidence that FNO can serve as a high-fidelity replacement for traditional FEM solvers within the reinforcement learning loop.

**State space complexity.** The state space in our construction environment scales exponentially with grid resolution. For a $20 \times 20 \times 4$ voxel grid with binary occupancy states, the theoretical state space encompasses $2^{1600} \approx 10^{482}$ possible configurations. When constrained to valid $4 \times 2 \times 1$ brick placements, each brick admits 2,584 distinct placement positions, yielding combinatorial complexity that renders exhaustive search intractable.

Our hierarchical agent architecture successfully identifies structurally sound, material-efficient solutions within this space. The key insight lies in the emergence of implicit construction heuristics not explicitly programmed into the reward function: **structural coherence** (avoiding unsupported elements without explicit geometric constraints), **load distribution** (utilizing symmetrical configurations to minimize stress concentrations).

### 6.4 Low-Level Executor Ablation

To isolate the contribution of WFC to overall performance, we replaced it with a trained PPO policy, while the high-level plan is produced by the same 2D planner described in Section 5.3. The low-level PPO agent is trained with a sparse reward signal: zero at all intermediate steps, and equal to the IoU between the high-level plan and its final realization at the terminal step. At each step, the agent places a single brick on the scene. When comparing executors, only the high-level planner is trained: the low-level PPO policy is used as a fixed pretrained checkpoint, while WFC is rule-based and requires no training.

The results are illustrated in Figure 8; all metrics are smoothed using an exponentially weighted moving average, the shaded regions show the corresponding minimum and maximum values across three seeds. As can be seen in the top plot, WFC achieves a higher and more stable IoU throughout the training. A similar trend holds in the bottom plot, where WFC maintains a higher IoU across varying brick counts, with lower variance for any fixed brick count. Consequently, the PPO-based executor introduces additional realization errors between the plan and the final construction. However, when our method uses a PPO-based executor, it quickly converges to a local optimum with a smaller brick count (middle plot). As a result, the produced structures do not satisfy the stress threshold, which reduces the effectiveness of the whole approach.

This ablation clarifies the role of WFC in the hierarchical system. WFC provides a more reliable mapping from high-level intent to executed construction. Such consistency enables the high-level planner to focus on strategic decisions rather than compensating for low-level executor errors.

### 6.5 Qualitative Analysis

To further elucidate the learning dynamics and qualitative outcomes of our approach, we provide visual examples of structures generated at different stages of training, focusing on stress distributions and structural characteristics.

During the initial training phase, the model frequently generated suboptimal configurations (Figure 9a), including designs with hanging or poorly supported bricks. These structures were unable to meet the allowable stress criteria, as evidenced by substantial stress concentrations that exceeded the material threshold.

As training progressed, the agent learned to construct more stable assemblies (Figure 9b). The frequency of failure cases decreased and the stress distribution became more uniform, although in certain regions local peaks in stress still appeared. These intermediate solutions often met the target height and connectivity requirements, but did not optimize material efficiency yet.

In the later stages of training, the model consistently produces high-quality structures (Figures 10a and 10b). These solutions not only conform to the target stress limits, but also significantly reduce the number of bricks required, demonstrating the efficiency of the learned material without sacrificing structural integrity. The stress gradients are smoother and critical regions are reinforced, reflecting advanced planning and a refined balance between stability and resource usage.

In general, these results illustrate a key property of our approach: the ability to autonomously discover design strategies that satisfy physical requirements and optimize resource consumption. The qualitative progression from failure cases to threshold-compliant solutions to resource-effective constructions demonstrates that the

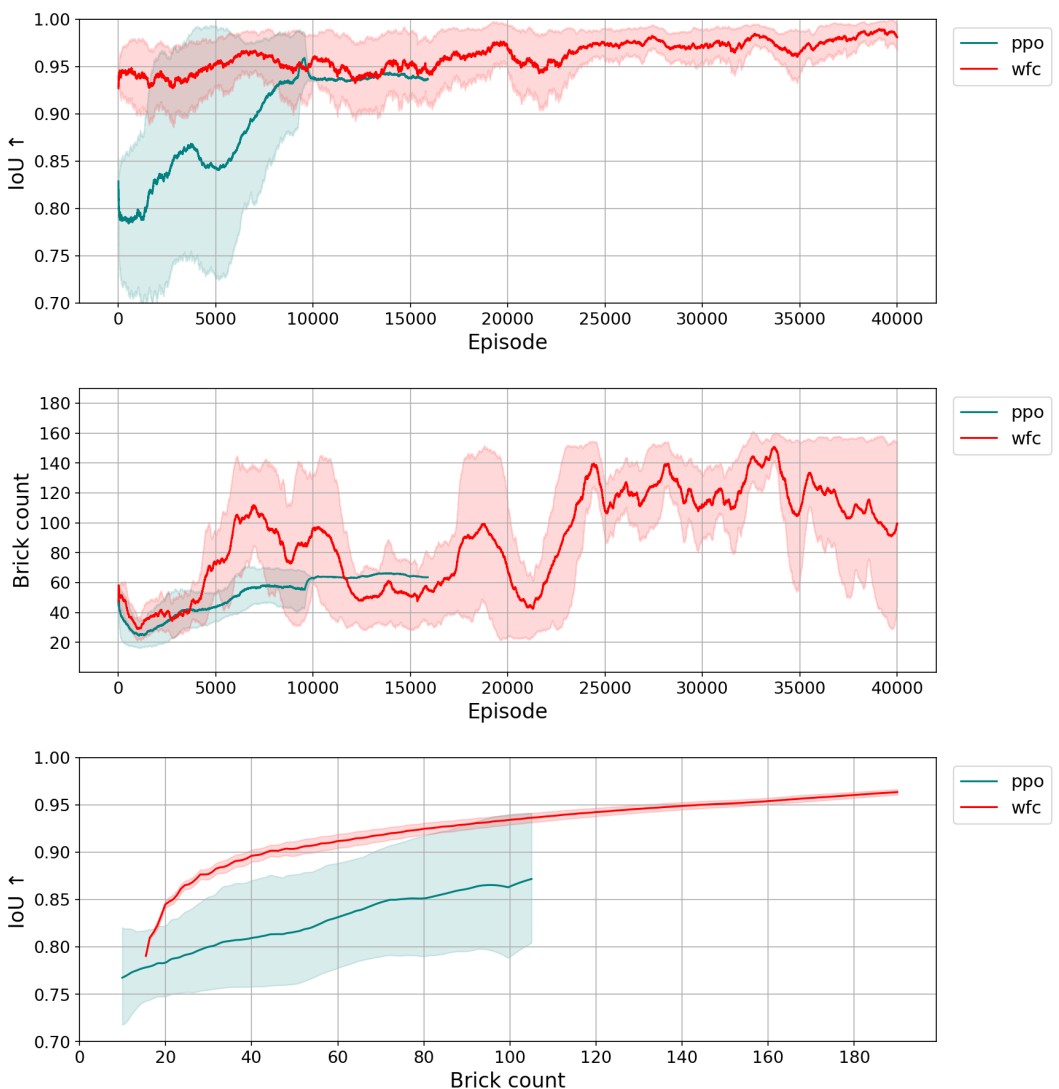

Figure 8: Comparison of low-level executors (high-level planner trained, executors frozen). *Top*: IoU between the high-level plan and its realized structure over episodes. *Middle*: Placed brick count over episodes. *Bottom*: IoU as a function of brick count. PPO runs are shorter because the high-level policy entropy reached zero, indicating early convergence to a local optimum.

proposed framework is capable of sophisticated, constraint-aware reasoning in complex structural assembly tasks.

Our results highlight that hierarchical decomposition is central to effective structural construction: separating strategic planning from tactical execution improves structural performance. The high-level planner can focus on global optimization, while the low-level executor ensures local constraint satisfaction. Despite the apparent simplicity of our discrete assembly task, the approach demonstrates effective navigation of a large solution space. The hierarchical learning architecture enables the agent to discover construction strategies and structural principles that extend beyond the explicitly encoded reward signals. Incorporating stress analysis directly into the reward function enables the learning of structurally sound construction strategies that would be difficult to discover through shape-based objectives alone, while the WFC-based executor provides efficient constraint handling, making real-time construction with complex constraint sets feasible.

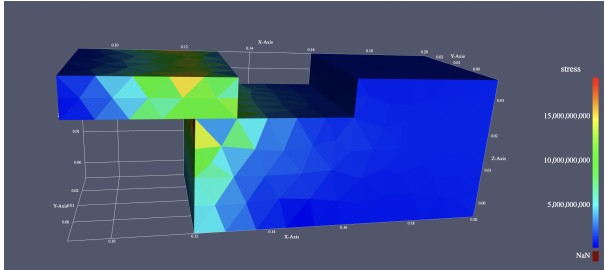
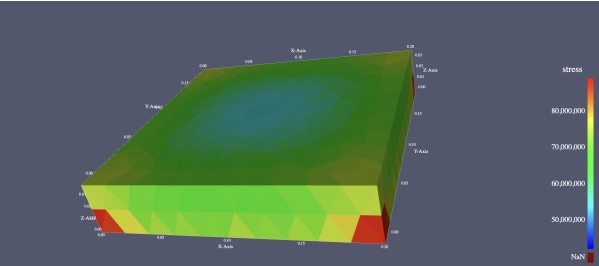

(a) Such designs often failed to satisfy the prescribed stress threshold and frequently contained overhanging elements with inadequate support.

(b) The stress is more evenly distributed, with the majority of the structure satisfying the threshold, though isolated high-stress regions remain.

Figure 9: Early and intermediate-stage structures generated during training. The left panel illustrates a failure case with high stress concentrations, while the right panel shows a partially stabilized structure where most stresses remain below the critical threshold, although some localized peaks persist.

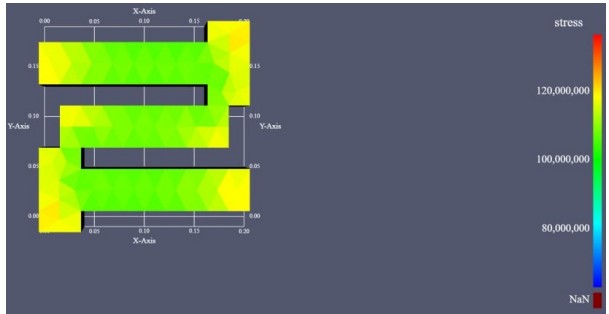
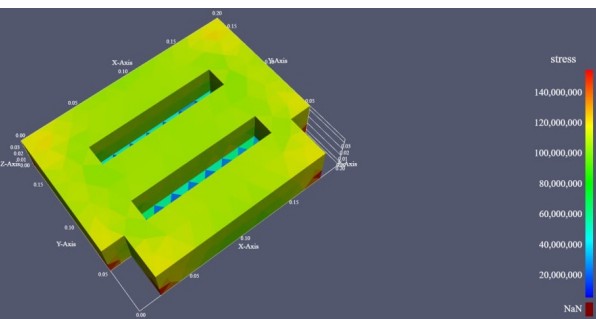

(a) Mature design after extended training. The structure is compact, uses fewer bricks, and successfully maintains stresses below the prescribed limit. This demonstrates the agent's ability to discover both efficient and physically valid solutions.

(b) Alternative top-down view of an optimized design at convergence. The agent has learned to maximize spatial efficiency while ensuring all stress regions remain safely below the failure threshold.

Figure 10: Mature structures achieved after extended training. Both designs meet the stress constraints while using significantly fewer blocks. The solutions demonstrate the agent's ability to optimize both material efficiency and structural reliability.

# 7 Discussion and Limitations

Our experiments cover five planner/solver/baseline combinations (2D+FEM, 3D+FEM, 3D+FNO, Random+WFC, and a non-hierarchical flat PPO) across three scene scales ($10 \times 10$, $15 \times 15$, $20 \times 20$). The target height is fixed to $H = 4$ in the main results; we treat this as a representative benchmark rather than a structural assumption of the method. The framework itself does not encode a specific target height, material, or loading condition. A different target height enters only through $h_{\text{target}}$ in Equation Height reward; a different material changes only the stress threshold $\sigma_{\text{thld}}$ and the elastic constants used by the FEM solver; a different loading scenario corresponds to a different right-hand side $\mathbf{f}(s)$ in the linear elasticity system. None of these substitutions touches the planner, the WFC executor, or the reward structure. The FNO-vs-FEM substitution in Section 5.2 follows the same pattern: the agent optimizes a different reward in the strict sense, since $\sigma(s)$ is replaced by $\hat{\sigma}(s)$, and still converges to comparable structures.

We do not include direct empirical comparisons with Brick-by-Brick (Chung et al., 2021) and BrECS (Ahn et al., 2024) because those methods solve a different subtask: they take a predefined target shape and reconstruct it from elementary bricks, whereas our task is to discover a stress-optimal shape in the first place. Running a shape-reconstruction baseline through our stress pipeline would primarily evaluate the quality of the supplied target rather than the reconstruction algorithm. Supplying these methods with a structurally

weak target would yield an accurate reconstruction that fails the stress test, while supplying a successful design would lead to a passing reconstruction by construction. The Random+WFC and flat PPO baselines we report instead share our problem setting.

Several limitations remain. The reward function is hand-designed: the weights $w_h$, $w_{\text{vol}}$ are set so that no failed structure can exceed the reward of a successful one (Section 5.1). Physics simulation introduces computational overhead, though asynchronous implementation and neural operator surrogates partially mitigate this impact. The current implementation uses simplified material models and loading conditions, with a planned extension to more complex structural analysis for future work. Finally, experiments focus on a single brick type; extension to multiple brick types and complex shapes is ongoing.

## 8    Conclusion

This study demonstrates that hierarchical decomposition, paired with physics-integrated rewards, enables agents to discover structures that satisfy stress constraints and minimize material, surpassing a 2D planning alternative that lacks full spatial reasoning. The low-level WFC executor complements the high-level policy by enforcing local feasibility and alignment with the global plan, supporting consistent success across runs and seeds. Moreover, an FNO-based surrogate effectively replaces FEM for reward computation without degrading ultimate performance, serving as a valuable proof-of-concept for integrating neural surrogates into physics-constrained design tasks.

## Acknowledgments

The Claude Sonnet model was used to verify grammar, vocabulary, and overall clarity throughout the manuscript.

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

## A  Notation

Table 1 summarizes the symbols that appear frequently throughout the paper.

| Symbol | Description |
|---|---|
| *Environment* | |
| $L, W, H$ | Dimensions of the fine construction grid (length, width, height) |
| $\tilde{L}, \tilde{W}$ | Dimensions of the coarse planning grid |
| $s$ | A candidate structure (binary occupancy grid) |
| $b$ | A brick (composite of $1 \times 1$ tiles) |
| $\Omega, \Omega^*$ | Set of valid structures, and feasible subset (stress and height satisfied) |
| *Objectives and reward* | |
| $\sigma_{vm}$ | Von Mises stress (engineering criterion) |
| $\sigma(s)$ | Maximum von Mises stress over all bricks in structure $s$ |
| $\sigma_{\text{thld}}$ | Stress threshold |
| $h, h_{\text{target}}$ | Height of the structure and target structure height |
| $N_{\text{brick}}$ | Number of bricks placed in the structure |
| $V_{\text{max}}$ | Maximum allowed number of bricks |
| $R$ | Total episode reward |
| $R_\sigma, R_h, R_{\text{vol}}$ | Stress, height, and volume reward components |
| $w_h, w_{\text{vol}}$ | Reward weights for the height and volume terms |
| *High-level planner* | |
| $\tilde{P}$ | Coarse plan produced by the planner |
| $P$ | Upsampled fine-grid plan |
| $k_L, k_W$ | Integer scaling factors between coarse and fine grids |
| $a_{\text{color}}$ | Block-type sub-action ($\{0, 1\}$) |
| $a_{\text{shape}}$ | Shape-index sub-action (from the shape library) |
| $a_{\text{center}}$ | Placement-center sub-action |

Table 1: Notation used throughout the paper.

## B  Implementation Details

The high-level PPO planner employs a 3D convolutional neural network (CNN) encoder to process the 3D grid-based environment. The hyperparameters for PPO training are provided in Table 3. The encoder consists

of a two-layer 3D CNN with ReLU activations: the first convolution layer maps from 1 input channel (binary occupancy) to 32 channels using a 3×3×3 kernel, followed by a second layer that transforms 32 channels to 16 output channels with the same kernel size. The padding and stride settings maintain the spatial resolution throughout the encoding. The encoder thus preserves spatial structure and encodes volumetric features relevant for planning, and its flattened output serves as the input for the actor-critic network detailed in Table 4, Table 5.

| Parameter | Value |
|---|---|
| coarse grid size | 10 |
| fine grid size | 20 |
| target height | 4 |
| num_envs | 20 |
| use augmentation | True |
| wfc greedy collapse | True |

Table 2: Main hyper-parameters.

| Parameter | Value |
|---|---|
| total_timesteps | 1,000,000 |
| buffer_size | 5000 |
| num_episodes_in_batch | 20 |
| anneal_lr | True |
| update_epochs | 10 |
| tau | 0.95 |
| norm_adv | True |
| clip_coef | 0.2 |
| clip_vloss | True |
| entropy_coeff | 0.0015 |
| critic_coeff | 0.5 |
| gradient_clip | 5.0 |
| target_kl | 0.01 |
| num_minibatches | 2 |
| learning_rate | 0.0002 |
| gamma | 0.999 |

Table 3: PPO hyper-parameters.

| Component | Architecture |
|---|---|
| Actor trunk | Linear($d \rightarrow 64$) + Tanh; Linear($64 \rightarrow 64$) + Tanh |
| Critic | Linear($d \rightarrow 64$) + Tanh; Linear($64 \rightarrow 64$) + Tanh; Linear($64 \rightarrow 1$) |
| Policy heads | Linear($64 \rightarrow |\mathcal{A}_{color}|$); Linear($64 \rightarrow |\mathcal{A}_{shape}|$); Linear($64 \rightarrow |\mathcal{A}_{center}|$) |

Table 4: Actor–critic architecture (2D planner). Here $d$ is the flattened encoder output dimension.

The training performance of the FNO, using the architecture and optimization hyperparameters detailed in Table 6, is illustrated in Figure 11. The plot omits the initial value to prevent a large drop in error (from $\sim 1.5$ to $\sim 0.15$) from compressing the y-axis and causing the subsequent convergence to appear negligible. The model validation loss converges rapidly, achieving a minimum of approximately 5% l2-error within the first 80 training epochs. This fast convergence leads to a model that accurately estimates the stress fields in a diverse validation set, which was constructed using stratified sampling to ensure a rich mix of geometric configurations and stress profiles.

| Component | Architecture |
|---|---|
| Actor trunk | Linear($d \to 256$) + ReLU; Linear($256 \to 256$) + ReLU |
| Critic | Linear($d \to 256$) + Tanh; Linear($256 \to 256$) + Tanh; Linear($256 \to 1$) |
| Color head | Linear($256 \to 64$)+ReLU; Linear($64 \to 64$)+ReLU; Linear($64 \to |\mathcal{A}_{color}|$) |
| Shape head | Linear($256 + 64 \to 64$)+Tanh; Linear($64 \to 64$)+Tanh; Linear($64 \to |\mathcal{A}_{shape}|$) |
| Center head | Linear($256 + 64 + |\mathcal{A}_{shape}| \to 128$)+Tanh; Linear($128 \to 128$)+Tanh; Linear($128 \to |\mathcal{A}_{center}|$) |

Table 5: Actor–critic architecture (3D planner).

| Parameter | Value / Setting |
|---|---|
| Input resolution | $64 \times 64 \times 12$ |
| Input channels | 1 (SDF) |
| FNO layers | 4 |
| Fourier modes (per spatial dim.) | 16 |
| Hidden channels | 32 |
| Projection channel ratio | 4 |
| Output channels | 1 (von Mises stress) |
| Optimizer | AdamW |
| Learning rate | $3 \times 10^{-4}$ |
| Weight decay | $1 \times 10^{-3}$ |
| LR schedule | Cosine annealing |
| Training epochs | 700 |

Table 6: Hyperparameters for the FNO-based physics approximator.

## C   WFC

The tile encoding for both orientations is illustrated in Figure 12, where each colored cell shows the tile code assigned to that spatial position within the brick. Code 0 denotes an empty space. Every tile defines adjacency rules that specify which tile codes are permitted on each of its four faces ($X^+$, $X^-$, $Y^+$, $Y^-$). Internal faces, which are edges shared between tiles of the same brick, accept exactly one tile code (the structurally required neighbor). For example, the upper-left corner of a vertical brick (code 1) requires code 2 on its $X^+$ face and code 3 on its $Y^-$ face. The set of codes accepted by an external face, which is exposed to other bricks or empty space, is broader. For example, code 1 with a free left edge $X^-$ accepts $\{0, 2, 4, 6, 8, 12, 16\}$, comprising empty space or any tile whose right edge is likewise external. Similarly, code 1 with a free top edge $Y^+$ accepts $\{0, 7, 8, 13, 14, 15, 16\}$. Code 0 accepts only tiles whose corresponding face is also external.

Each brick-level rule computes a coefficient in $\{\epsilon, 1\}$, then the weight of tile is scaled by $\min(1, Consistent(b), Solid(b), Based(b))$. $Solid(b)$ iterates over brick tiles and checks whether the required adjacency face exists in the current cell's connection map. If any tile position falls outside valid boundaries of the grid, $Solid(b) = \epsilon$; otherwise $Solid(b) = 1$. The brick is considered supported ($Based(b) = 1$) if at least one downward neighbor is occupied; otherwise $Based(b) = \epsilon$.

$Consistent(b)$ traverses the brick's tile map and checks two conditions per neighbor cell. First, if the neighbor has already collapsed to tile code $t'$ and $t' \neq t_{\text{expected}}$, the rule immediately returns $\epsilon$. Second, if the neighbor is not collapsed but the expected tile code is no longer in its valid tile set, the rule also returns $\epsilon$. If neither hard conflict is triggered, a soft penalty is applied for each tile of $b$ that falls outside the plan mask. Let $k$ denote the number of such out-of-plan tiles, then the penalty multiplier is applied sequentially as:

$$\gamma_k = \begin{cases} 0.9 & \text{if } k \leq 4 \\ \dfrac{0.9}{k^2} & \text{if } k > 4 \end{cases}$$

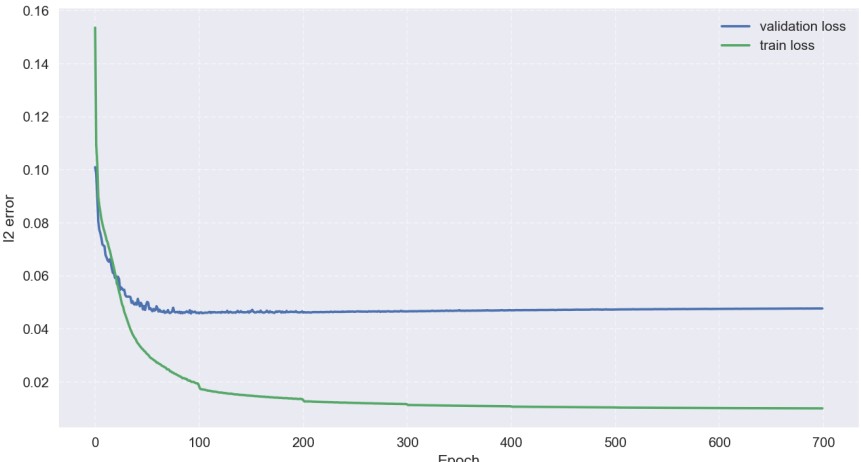

Figure 11: Convergence of the FNO for stress prediction.

| 0 | 0 | 0 | 0 | 0 | 0 |
|---|---|---|---|---|---|
| 1 | 2 | 0 | 0 | 0 | 0 |
| 3 | 4 | 0 | 0 | 0 | 0 |
| 5 | 6 | 9 | 10 | 11 | 12 |
| 7 | 8 | 13 | 14 | 15 | 16 |

Figure 12: Tile encoding scheme

