# OpenReview forum: "Hierarchical Multi-Level 3D Geometry Generation with Stress-Aware Learning"
_TMLR — Accepted by TMLR_

### Review · Reviewer_UH67 · 2025-12-14

**Summary Of Contributions:**

This paper proposes a hierarchical, physics-aware framework for 3D LEGO-style structural generation that explicitly optimizes for structural integrity under stress, rather than geometric similarity alone. The claimed key contributions are as follows: 1. Stress-aware formulation of 3D combinatorial construction 2. Hierarchical planning architecture w/ PPO + WFC with some empirical experiments.

**Audience:**

Yes

**Audience Explanation:**

Target audiences include:
1. researchers in reinforcement learning for sequential decision-making.
2. researchers in physics-informed learning.

**Claims And Evidence:**

No

**Claims Explanation:**

I doubt this paper is heavily relying on LLM-generated text. Several evidence:
1. There are too many sections, and way too many bullet lists, polished and formulaic. There are noticeable, unusually wide gaps between bullet points that resemble LLM-generated LaTeX code behavior.
2. More importantly, many key notations, methodologies are not introduced, at least including:
- In abstract, FEM and FNO were not explained with full names.
- 3.1 should be presented with at least some figures as examples for readers to understand their setup, given than 3D figures are not that hard to visualize.
- 3.2 $\sigma$ is not defined in the primary objective for its first occurence in the paper.
- magic number $w_h = 0.3$. Based on the paper logic, why not choosing 0.2 or 0.4 so that the total reward is still below 1?
- doesn't give any example of $\sigma_{thld}$ for different types of brick material
- $w_\text{vol}$ shouldn't be volume reward but it was introduced together as $R_\text{vol}$
- "The extracted features are flattened and .... placement coordinates.": authors introduced fully connected actor layers without introducing any architectural details or pointers to appendix.
- Authors didn't include PPO algorithm's discussion *in their specific problem setup*, even if it's a well-known RL algorithm.
3. Potentially hallucinated citations:
- [6]'s authors should be Zhang, W.; Ni, P.; Zhao, M.; Du, X.
- [11]'s authors should be Minwoo Shin, Minjee Seo, Hayoung Choi, Jaemin Jung, Kyungho Yoon, all first names were incorrect.

**Requested Changes:**

Given that there are too many claims (see above) made in the paper unjustified, I do not believe the paper is ready for acceptance in its current form.

---

> ### Comment · Action_Editor_27ME · 2026-04-20
>
> Dear Reviewer UH67,
>
> Could you please review the authors’ responses and submit your final recommendation? Thank you again for your time and effort in reviewing this paper.
>
> Best regards,
> AEs

---

### Review · Reviewer_14oQ · 2026-01-26

**Summary Of Contributions:**

The paper studies a stress aware 3D LEGO style construction task where the goal is not shape matching (like prior work, viz. Brick-by-brick) but producing a stable structure that reaches a target height while minimizing peak von Mises stress. It uses a hierarchical Reinforcement Learning (RL) setup where a PPO high level planner proposes coarse fill regions and a low level Wave Function Collapse (WFC) executor realizes them at brick resolution. It introduces a physics-based shaped reward signal that is efficiently evaluated in the learning loop using a trained Fourier Neural Operator (FNO) over a slow Finite Element Methods (FEM) scheme.

### Strengths:

- The problem formulation is a nice shift from IoU driven construction toward physically meaningful objectives.
- The reward design explicitly targets stress threshold satisfaction first and then adds a material usage bonus (using $N_\text{bricks}$), which matches the stated goals.
- Using an operator surrogate for stress prediction is a reasonable way to make physics in the loop practical, and the paper reports large speedups over FEM while keeping downstream learning behavior similar. Shows promise of getting around expensive FEM simulations to solve a larger variety of problems using RL and simulators.

### Weaknesses:

- The rewards are primarily hand-engineered and it appears very specific to this problem. Experiments are limited in scope (single target height of 4, single map size) and it is hard to say how far the approach generalizes as well.
- Several interface details between levels are underspecified, which makes it harder to reproduce or understand failure modes. Examples include how the 10×10×4 coarse plan maps to the 20×20×4 executor frames, how the frame size $F$ is chosen, and what feasibility means in practice beyond action masking and WFC heuristics.
- The computational claims are hard to evaluate because the paper gives relative speedups but not wall clock timing in the actual training loop, nor clear hardware and resource accounting for FEM service and FNO training. Since reward computation is asynchronous, it is also unclear how much of the training time is actually spent on the solver versus environment throughput.

**Audience:**

Yes

**Audience Explanation:**

- Readers interested in combining RL with physics based objectives and surrogate modeling may find the pipeline useful, especially the use of an operator model as a drop in reward evaluator. The WFC executor may also be of interest to people who care about constraint satisfaction in discrete generation.
- The work might be less broadly interesting to the general ML community since much of the contribution is in integrating known components rather than introducing a new learning algorithm. The novelty is more in the task objective and system design than in new theory or methodology.

**Broader Impact Concerns:**

No obvious ethical concerns.

**Claims And Evidence:**

Yes

**Claims Explanation:**

- The problem formulation is novel and the feedback loop to the RL agent is run in an efficient manner using FNOs over a full FEM based evaluation. The claim that an FNO can replace FEM for rewards is supported by similar end performance between the FEM guided and FNO guided 3D agents, plus qualitative stress field comparisons.
- The main claims about hierarchical planning are supported by the 2D versus 3D comparison, where the 3D planner achieves much higher success and lower stress to threshold ratios. Still, the evidence would be stronger with a broader set of baselines and more varied task settings.

**Requested Changes:**

1. (Important) Please add more to WFC executor details and clean up presentation. Some concrete additions would be, specify the coarse to fine mapping, the WFC frame size $F$, how $a_{shape}$ and its library are constructed, and how support and connectivity constraints are enforced at each level. Lastly please unify plot formatting (Fig. 4 and Fig. 5) and give valid ranges for hyperparameters like $w_h$ and $\eta$ (e.g. $\in \mathbf{R}^+ > 0$).  If the authors intended high-level frames to align with coarse cells then $F=2$ is plausible, but this should be stated explicitly.
2. (Important) Please report end to end compute and timing. Also include wall clock per episode or per PPO iteration for FEM versus FNO, the hardware and cloud setup used, as well as the one time cost to train the FNO that enables the speedup downstream.
3. (Important) Please cite additional related work such as the original FNO papers:

    a. Fourier Neural Operator for Parametric Partial Differential Equations, Li et. al

    b. Neural Operator: Learning Maps Between Function Spaces With Applications to PDEs, Kovachki et. al

4. (Nice to have) A deeper treatment of the background such as short intuition paragraph for WFC would help readers without a graphics background. A start could be describing it as maintaining a set of allowed brick placements per cell and repeatedly committing to a choice while propagating local compatibility constraints.
5. (Nice to have) Experiments over a wider range of target heights or additional setups showing some generalizability of the approach (say following a similar setting as Brick-by-brick) and the hand-crafted reward.
6. (Nice to have) $a_\text{shape}$ is sampled from a finite set of regions. Reasoning as to how a user should structure this based on the problem would be helpful. For example, 8x2 is omitted for the two brick case and it is unclear why.

---

### Review · Reviewer_eGFY · 2026-03-09

**Summary Of Contributions:**

The paper introduces a hierarchical reinforcement learning framework designed for assembling discrete 3D building elements, akin to LEGO brick construction. Unlike prior approaches that optimize primarily for shape similarity metrics like Intersection over Union (IoU), this method optimizes for structural stability, load distribution, and material efficiency. The proposed architecture utilizes a two-level agent system:
* A high-level Proximal Policy Optimization (PPO) planner generates coarse 3D layouts.
* A low-level Wave Function Collapse (WFC) executor enforces strict local placement and support constraints.
To compute the physics-aware reward, the authors employ a Finite Element Method (FEM) solver to evaluate the structure's maximum von Mises stress. To mitigate computational bottlenecks, they also demonstrate the viability of replacing the FEM solver with a Fast Fourier Neural Operator (FNO) surrogate model.

**Strengths**

Novel and Practical Formulation: Shifting the primary optimization objective from geometric shape reconstruction to physics-aware structural integrity is a highly practical advancement for automated manufacturing and robotics.

Elegant Hierarchical Design: The decoupling of global strategic planning from local constraint satisfaction effectively navigates a massive combinatorial state space. The grouping mechanism within the WFC algorithm is a particularly clever way to evaluate support constraints at the brick level rather than the cell level.

Integration of Surrogate Modeling: Training a 3D FNO to predict the von Mises stress field from a single-channel Signed Distance Function (SDF) representation is a strong application of modern neural operators. The FNO achieves an impressive 85x to 150x speedup over conventional FEM during batch inference.

**Weaknesses**

Generalizability Limitations: The current empirical validation is constrained to a single brick type and simplified, uniform loading conditions. Expanding the environment to handle multiple brick shapes, dynamic loading scenarios, or different material yield strengths (e.g., ABS plastic versus aluminum) would significantly strengthen the paper's claims.

Critically Missing Baselines: The authors explicitly motivate their method by critiquing prior geometric models (like Brick-by-Brick and BrECS) for failing to build stable structures. However, they only evaluate their proposed method against an ablated 2D version of itself. Failing to run a purely geometric baseline through their stress-evaluation pipeline leaves their primary claim untested against the state-of-the-art.

**Audience:**

Yes

**Audience Explanation:**

Yes, the findings of this paper will certainly be of interest to several sub-communities within the TMLR audience.

Physics-Informed Generative AI (AI for Science): There is a rapidly growing subfield focused on embedding physical laws into neural network architectures. The authors' use of a Fast Fourier Neural Operator (FNO) to approximate Finite Element Method (FEM) stress field, and integrating that surrogate directly into the RL reward loop, serves as a highly relevant case study for researchers working with physics-informed neural networks and neural operators.

Robotics and Automated Design: Existing literature on 3D discrete assembly often relies heavily on geometric metrics, such as maximizing Intersection over Union (IoU) with a target shape. The shift proposed in this paper—formulating the problem as a multi-objective optimization that explicitly targets physical stability (von Mises stress) and resource efficiency, addresses a crucial gap between visually plausible generation and physically viable construction. This will appeal directly to researchers at the intersection of ML, robotics, and architectural design.

**Broader Impact Concerns:**

-

**Claims And Evidence:**

No

**Claims Explanation:**

While the authors provide strong evidence for the internal mechanics of their proposed architecture, the broader claims regarding performance against the state-of-the-art and system-level computational efficiency lack sufficient and accurate supporting evidence.

Missing State-of-the-Art Baseline Comparisons: The authors explicitly motivate their work by critiquing existing geometry-based assembly methods—specifically citing Brick-by-Brick and BrECS, arguing that optimizing for geometric metrics like IoU does "not ensure that the resulting structure can support its own weight or resist external forces". However, the paper's experiments only compare the proposed 3D planner against an ablated 2D version of itself. They do not evaluate a purely geometric baseline (like Brick-by-Brick) through their FEM stress pipeline to actually prove that prior methods fail at this task. Without this direct comparison, the claim that stress-aware optimization produces physically superior structures compared to existing state-of-the-art models remains an untested hypothesis rather than a proven fact.

Misleading Claims Regarding FNO Computational Speedup: The submission claims that replacing the traditional FEM solver with an FNO surrogate "overcomes" computational bottlenecks, enabling "efficient and scalable training". The text cites an 85x to 150x speedup over conventional FEM. However, as revealed by the authors in the OpenReview rebuttal, the end-to-end training time for 17k iterations was actually longer with the FNO (48.7 hours) than with the FEM solver (44.2 hours). The authors attribute this to CPU-GPU transfer overhead and note that physics computation only accounts for ~18% of the per-iteration time. Therefore, the claim that the FNO solves the system-level computational bottleneck for RL training is currently unsupported by the actual wall-clock metrics.

Well-Supported Internal Ablations: To the authors' credit, the claims that 3D spatial reasoning is strictly necessary over 2D reasoning for structural stability are very clearly supported by the ablation studies shown in Figure 7. Similarly, the claim that the agent learns to balance material efficiency with stability is well-evidenced by the quantitative cubic count metrics and qualitative visualizations.

**Requested Changes:**

Critical Requirement for Acceptance: While the introduction of a physics-aware reward function is a highly commendable and practical direction, the current experimental section suffers from a critical "straw man" evaluation. The submission explicitly critiques state-of-the-art RL assembly methods, specifically Brick-by-Brick (Chung et al., 2021) and BrECS (Ahn et al., 2024), arguing that their reliance on geometric shape similarity (IoU) fails to ensure structural integrity or optimal load distribution.

Minor:
Please use parenthetical citation \citep when citing in a sentence.
For the figures, prefer using vector graphics (e.g. the pdf) or use higher resolution png files because currently the figures are slightly pixelated (e.g. figure 3)
Figure 4: Convergence of the FNO for stress prediction. - I don’t think this plot belongs to the main text. Presenting just the test performance should be sufficient and move the figure to the appendix.

---

> ### Author Response · Authors · 2026-03-11
> **Response to Reviewer eGFY**
>
> We sincerely appreciate your feedback. You can find our new revision in the submission page. In the following, we will answer your concerns and questions.
>
> > Missing State-of-the-Art Baseline Comparisons: The authors explicitly motivate their work by critiquing existing geometry-based assembly methods—specifically citing Brick-by-Brick and BrECS, arguing that optimizing for geometric metrics like IoU does "not ensure that the resulting structure can support its own weight or resist external forces". However, the paper's experiments only compare the proposed 3D planner against an ablated 2D version of itself. They do not evaluate a purely geometric baseline (like Brick-by-Brick) through their FEM stress pipeline to actually prove that prior methods fail at this task. Without this direct comparison, the claim that stress-aware optimization produces physically superior structures compared to existing state-of-the-art models remains an untested hypothesis rather than a proven fact.
>
> > Critical Requirement for Acceptance: While the introduction of a physics-aware reward function is a highly commendable and practical direction, the current experimental section suffers from a critical "straw man" evaluation. The submission explicitly critiques state-of-the-art RL assembly methods, specifically Brick-by-Brick (Chung et al., 2021) and BrECS (Ahn et al., 2024), arguing that their reliance on geometric shape similarity (IoU) fails to ensure structural integrity or optimal load distribution.
>
> We thank the reviewer for highlighting this point and agree that demonstrating the physical advantages of stress-aware optimization is critical. However, we omitted direct empirical comparisons because the problem formulation in our work differs from those in Brick-by-Brick and BrECS.
>
> In our case, the task can be divided into two subtasks: finding a stress-optimal structure, and constructing the found structure from elementary bricks. Brick-by-Brick and BrECS excellently solve the second subtask, but they do not address the first. If we provided them with a structurally weak target shape (example: Figure 7a, p. 11), they would accurately reconstruct it but fail under our FEM stress metrics. Conversely, if we select a successful design as the target (example: Figure 7b, p. 11), the constructed result will likely be successful too.
>
> As a result, running these baselines through our stress pipeline would evaluate the properties of the arbitrary target shape we chose, rather than the efficiency of the baseline algorithms. Instead of an unfair empirical comparison, we decided to contrast our problem formulation (multiobjective structural optimization) against prior work (shape similarity):
> - Introduction (p. 1): “... current approaches to LEGO brick construction focus on replicating a target shape …”
> - Related Work (p. 2): ”Although this approach showed improvements in assembly speed and constraint handling, it remained focused on shape completion rather than structural optimization.”
> - Optimization Objective and Reward (p. 3): ”Unlike previous work Chung et al. (2021); Ahn et al. (2024) that optimizes for shape similarity, we formulate construction as a multiobjective optimization problem.”
>
> We hope this clarifies why direct empirical comparisons with these baselines would not be informative in this setting.
>
> > Minor: Please use parenthetical citation \citep when citing in a sentence. For the figures, prefer using vector graphics (e.g. the pdf) or use higher resolution png files because currently the figures are slightly pixelated (e.g. figure 3) Figure 4: Convergence of the FNO for stress prediction. - I don’t think this plot belongs to the main text. Presenting just the test performance should be sufficient and move the figure to the appendix.
>
> The citations have been updated to use \citep and \citet. The format of Figure 3 has been changed from PNG to PDF to improve image quality. Additionally, Figure 4 and its corresponding paragraph have been moved to the appendix to avoid cluttering the main text with excessive details.

---

> ### Author Response · Authors · 2026-03-11
> **Response to Reviewer eGFY (continuation)**
>
> > Misleading Claims Regarding FNO Computational Speedup: The submission claims that replacing the traditional FEM solver with an FNO surrogate "overcomes" computational bottlenecks, enabling "efficient and scalable training". The text cites an 85x to 150x speedup over conventional FEM. However, as revealed by the authors in the OpenReview rebuttal, the end-to-end training time for 17k iterations was actually longer with the FNO (48.7 hours) than with the FEM solver (44.2 hours). The authors attribute this to CPU-GPU transfer overhead and note that physics computation only accounts for ~18% of the per-iteration time. Therefore, the claim that the FNO solves the system-level computational bottleneck for RL training is currently unsupported by the actual wall-clock metrics.
>
> We agree that our claim regarding the FNO speedup may appear misleading without additional context. The reported 85×–150× speedup strictly refers to the isolated execution time of the FNO surrogate compared to the FEM solver for a single physics step, rather than the wall-clock time of the entire training loop.
>
> To provide context, the physics simulation was the primary bottleneck of our pipeline during early development. Because of this, we pursued two independent approaches: optimizing the traditional FEM solver (via dolfinx) and replacing dolfinx with a neural operator. The FEM optimization was faster to implement and achieved sufficient speedup; thus, the main experiments for our method were conducted using this implementation. By the time the FNO was trained, the urgent need to speed up the pipeline had already been resolved. Therefore, we were mostly interested in seeing whether the algorithm would converge to the same solution if the FEM solver was replaced by the operator. We then conducted isolated benchmarking, which yielded the 85x to 150x speedup results. Since this aligned with our expectations and previous experience with FNOs, we overlooked the timing discrepancy in the end-to-end scenario.
>
> To accurately reflect these findings, we have updated the manuscript. Specifically, we clarified the isolated nature of the 85x–150x speedup and reframed FNO integration as a proof-of-concept.
>
> **Abstract**
>
> Before: "We also show that replacing the computationally expensive finite element method solver with a fast Fourier neural operator achieves comparable performance, confirming the approach’s scalability for large-scale problems."
>
> After (p. 1): “We also show that replacing the finite element method solver with a Fourier neural operator achieves comparable performance, providing proof-of-concept that the proposed approach can work with neural surrogates.”
>
> **Section 4.3 Physics Integration**
>
> Before: "However, a major computational bottleneck arises from relying solely on traditional FEM solvers for the reward calculation in a reinforcement learning loop. To overcome this, we implement a physics approximator based on FNOs, which provides rapid and accurate stress field predictions from geometric data, enabling efficient and scalable training."
>
> After (p. 7): “However, relying solely on traditional FEM solvers for the reward calculation in a reinforcement learning loop can introduce computational bottlenecks. To explore potential solutions, we implement a physics approximator based on FNOs, which provides rapid and accurate stress field predictions from geometric data.”
>
> **Section 4.3 Physics Integration**
>
> Before: "This approach enables the integration of realistic physics into reinforcement learning for structural assembly, where computational efficiency is crucial without compromising fidelity. It allows the RL agent to learn to build minimal-stress 3D structures from bricks efficiently."
>
> After (p. 7): “This approach enables the integration of realistic physics into RL for structural assembly without compromising fidelity. It demonstrates that the RL agent can learn to build minimal-stress 3D structures from bricks using a neural surrogate.”
>
> **Section 5.2 Neural Operator Training**
>
> Before: "The FNO approximator achieves an 85× to 150× speedup over conventional FEM while maintaining efficient memory usage for large-scale 3D geometric inputs."
>
> After (p. 8): “When evaluated as a separate component on identical scenes, the FNO approximator achieves a 85× to 150× speedup over conventional FEM. However, when measuring the end-to-end iteration time, FEM and FNO yield comparable results, since the training pipeline was originally optimized for CPU-based FEM, and the physics computation comprises only 18\% of the per-iteration time.”

---

> ### Author Response · Authors · 2026-03-11
> **Response to Reviewer eGFY (continuation)**
>
> **Section 5.3 Structural Performance Results**
>
> Before: "Furthermore, they provide strong evidence that FNO can serve as a high fidelity, computationally efficient substitute for traditional FEM solvers within the reinforcement learning loop, enabling effective and scalable training for complex, physics-constrained tasks."
>
> After (p. 10): “Furthermore, they provide strong evidence that FNO can serve as a high-fidelity replacement for traditional FEM solvers within the reinforcement learning loop.”
>
> **Section 6 Conclusion**
>
> Before: "Moreover, an FNO-based surrogate effectively replaces FEM for reward computation without degrading ultimate performance, offering a practical path to scalable training in physics-constrained design tasks."
>
> After (p. 12): “Moreover, an FNO-based surrogate effectively replaces FEM for reward computation without degrading ultimate performance, serving as a valuable proof-of-concept for integrating neural surrogates into physics-constrained design tasks.”

---

> > ### Comment · Reviewer_eGFY · 2026-03-28
> >
> > Thank you for the detailed and thoughtful rebuttal, and for the extensive revisions made to the manuscript.
> >
> > Regarding the FNO Computational Claims: I deeply appreciate the transparency regarding the development timeline of the FEM and FNO pipelines. The systematic revisions made across the Abstract, Sections 4.3, 5.2, 5.3, and the Conclusion perfectly address my concerns. Reframing the FNO integration as a "proof-of-concept" accurately reflects the empirical reality of the current training loop while still highlighting the impressive isolated speedup of the surrogate model. I consider this critique fully and satisfactorily resolved.
> >
> > Regarding the Baseline Comparisons:
> > I acknowledge your argument regarding Brick-by-Brick and BrECS. Your point is well-taken: because those methods require a predefined target shape (shape imitation) rather than discovering a shape from scratch (shape discovery), passing their outputs through a stress-pipeline would primarily evaluate the quality of the provided blueprint rather than the algorithm itself. I agree that a direct 1:1 comparison with those specific methods is structurally incompatible.
> >
> > However, recognizing that you are tackling a novel subtask ("finding a stress-optimal structure") does not eliminate the need for an external baseline. Currently, the proposed 3D RL agent is only shown to be superior to a crippled (2D) version of itself. To confidently assess the efficacy of the learned policy, it must be compared to a non-RL or naive approach.
> >
> > To bridge this gap, I strongly encourage the inclusion of a simple heuristic or random baseline (e.g., a "Random + WFC" agent that randomly samples valid block placements, or a basic rule-based planner that stacks uniformly) evaluated on the same stress metrics and brick counts. Showing that the PPO agent's learned structural heuristics vastly outperform a naive 3D assembly strategy would provide the necessary external validation to anchor the paper's core claims.

---

> > > ### Author Response · Authors · 2026-04-03
> > > **Response to Reviewer eGFY**
> > >
> > > > To bridge this gap, I strongly encourage the inclusion of a simple heuristic or random baseline (e.g., a "Random + WFC" agent that randomly samples valid block placements, or a basic rule-based planner that stacks uniformly) evaluated on the same stress metrics and brick counts. Showing that the PPO agent's learned structural heuristics vastly outperform a naive 3D assembly strategy would provide the necessary external validation to anchor the paper's core claims.
> > >
> > > Thank you for this constructive suggestion. In response, we added a ‘Random + WFC’ baseline agent and evaluated it using the same metrics (stress, cubic count, and success rate), shown in Figure 6 in yellow. Section 5.3 was updated accordingly.
> > >
> > > **Section 5.3 Structural Performance Results**
> > >
> > > Before:
> > > Figure 6 presents ... illustrating the performance variance.
> > >
> > > After (p. 9):
> > > To confidently evaluate the effectiveness of our method, we decided to compare it with a naive approach: the high-level planner uses a random strategy with action masking, while the low-level execution is still handled by WFC. Figure 6 presents ... illustrating the performance variance. The random baseline was evaluated in a single run, as it shows nearly identical stable performance in 4,000 iterations.
> > >
> > > Before:
> > > As illustrated, both 3D planners successfully learn policies that reduce the stress-to-threshold ratio below 1.0, whereas the 2D planner consistently fails to do so.
> > >
> > > After (p. 9):
> > > As illustrated, the baseline exhibits the worst performance, fluctuating around a value of 10,000. The 2D planner only manages to drop below 100 toward the end of training, whereas the 3D planners successfully learn policies that reduce the stress-to-threshold ratio below 1.0.
> > >
> > > Before:
> > > … underscoring its instability and limited effectiveness.
> > >
> > > After (p. 9):
> > > … underscoring its instability and limited effectiveness. The naive baseline remains nearly constant at a success rate of roughly 17%, showing that perfect performance requires learned structural heuristics rather than relying on valid random placements.
> > >
> > > Before:
> > > The plot shows that all agents initially increase the brick count to expand the load‑bearing area at the top layer, which lowers the stress‑to‑threshold ratio by distributing loads more evenly.
> > >
> > > After (p. 10):
> > > The plot shows that all agents initially increase the brick count to expand the load‑bearing area at the top layer, which lowers the stress‑to‑threshold ratio by distributing loads more evenly. In contrast, the Random+WFC baseline maintains an almost flat brick count of roughly 70 bricks throughout the training.

---

### Author Response · Authors · 2026-02-04
**Response to Reviewer 14oQ**

We sincerely appreciate your feedback. You can find our new revision in the submission page. In the following, we will answer your concerns and questions.

> Please add more to WFC executor details and clean up presentation … how support and connectivity constraints are enforced at each level … A deeper treatment of the background such as short intuition paragraph for WFC would help readers without a graphics background. A start could be describing it as maintaining a set of allowed brick placements per cell and repeatedly committing to a choice while propagating local compatibility constraints

We have revised Section 4.2 ("Low-Level WFC Executor"). Specifically, we added:
1) A description of how the support constraint is enforced via tile grouping.
2) A brief introduction to WFC that describes it as collapse-propagation cycle.
3) A note that structural connectivity is ensured by restricting the WFC collapse operation to cells adjacent to the existing assembly.

> specify the coarse to fine mapping … how $a_{shape}$ and its library are constructed

We have added explanations for both points in Section 4.1 "High-Level Planner":
1) coarse to fine mapping (p. 5): $P_{x,y,z} = \tilde{P}_{\lfloor x/k_L \rfloor, \lfloor y/k_W \rfloor, z}$,
where $k_L = \frac{L}{\tilde{L}}$ and $k_W = \frac{W}{\tilde{W}}$ are integer scaling factors.
2) shape library construction (p. 6): we now describe the requirements to shape (it must be rectangular, it must fit within a coarse grid, and it is made up of several bricks).

> unify plot formatting (Fig. 4 and Fig. 5)

We have modified Figures 5 and 6 to match the styling of other figures. Specifically, the background color was changed to white, all fonts were set to black, and the font size was increased.

> give valid ranges for hyperparameters like $w_h$ and $\eta$

We have added the valid ranges for hyperparameters in Section 3.2 (p. 4). For example, $w_h \in (0, 0.5]$ and $\eta \in [1, \infty)$.

> Please report end to end compute and timing. Also include wall clock per episode or per PPO iteration for FEM versus FNO, the hardware and cloud setup used, as well as the one time cost to train the FNO that enables the speedup downstream.

We report wall-clock training times on a single NVIDIA Tesla T4 GPU:
- 3D + FEM (39k iterations) – 96h, 8.86s/iter
- 3D + FEM (17k iterations) – 44.2h, 9.3s/iter
- 3D + FNO (17k iterations) – 48.7h, 10.3s/iter

However, the FNO achieves 85×–150× speedup over FEM when evaluated in isolation on identical scenes (GPU-to-GPU, batch inference).

The observed end-to-end training time with FNO is comparable to FEM due to following reasons:
- The training pipeline was originally optimized for CPU-based FEM; integrating FNO introduced additional CPU-GPU memory transfer overhead.
- Physics computation comprises only ~18% of per-iteration time. The remaining 82% are: sampling by the scheduler, execution of the WFC algorithm, data preprocessing, PPO updates.

Nevertheless, we include the FNO results primarily to demonstrate generalizability: the hierarchical RL framework is agnostic to the physics solver type (FEM or approximation).

> Please cite additional related work such as the original FNO papers

We have added citations to the original FNO papers in the Related Work section.

>Experiments over a wider range of target heights or additional setups showing some generalizability of the approach (say following a similar setting as Brick-by-brick) and the hand-crafted reward.

By using an approximator instead of an FEM solver, we effectively optimize a different reward function, since $\sigma(s)$ is replaced by $\hat{\sigma}(s)$. Sections 5.3 and 5.4 show that both approaches yield similar quantitative and qualitative results. This example demonstrates the generalizability of the approach and the reward formulation, since any other type of loading condition (for example, loads applied to both the top and side surfaces) corresponds to the substitution of $\sigma(s)$ with $\hat{\sigma}(s)$.

---

### Decision · Action_Editor_27ME · 2026-05-13

**Recommendation:** Accept with minor revision

**Additional Comments:**

The paper proposes a hierarchical framework for stress-aware 3D modular structure generation using PPO, WFC, and physics-based stress evaluation. Unlike prior work focused on geometric similarity, the method explicitly optimizes structural stability and material efficiency. The submission is reviewed by three experts, who have actively engaged in a long discussion with the authors.

- Reviewer 14oQ found the revised paper improved after the rebuttal, with concerns about WFC details, timing analysis, and FNO claims largely addressed through additional pseudocode, experiments, and clarification of the FNO results. The reviewer still noted that the overall scope remains somewhat limited.

- Reviewer eGFY initially raised concerns about missing external baselines and unsupported FNO speedup claims. The later revision addressed these concerns by adding detailed timing analysis and the requested Random+WFC baseline (which is added after the final recommendation).

- Reviewer UH67 raised concerns about missing methodological details, missing ablations, and presentation quality. The later revision addressed many of these points through additional pseudocode, flat PPO and PPO-executor ablations, new figures, and substantial restructuring of the manuscript. (Some concerns are responded to after the final recommendation)

Based on the above discussion, I recommend acceptance. I also encourage the authors to further strengthen the final version by incorporating the remaining suggestions regarding experimental breadth, presentation clarity, and additional discussion of limitations.

**Audience:**

Yes

**Audience Explanation:**

The paper studies an interesting combination of hierarchical reinforcement learning, constraint-based generation, and physics-aware optimization for modular 3D assembly. While the contribution is primarily system-oriented rather than algorithmically novel, the problem setup and empirical findings should still be of interest to parts of the TMLR audience.

**Claims And Evidence:**

Yes

**Claims Explanation:**

The revised manuscript provides much stronger support for its main claims through additional baselines, ablations, and implementation details. In particular, the added Random+WFC baseline, flat PPO baseline, and PPO executor ablation help demonstrate the benefit of the hierarchical PPO+WFC design over simpler alternatives under the proposed stress-aware objective. Although the experimental scope is still somewhat limited and stronger external baselines would improve the paper further, the main claims are now supported by reasonably clear evidence.

---

> ### Author Response · Authors · 2026-06-03
> **Response to Action Editor**
>
> Dear Action Editor,
>
> We thank you for the careful evaluation and for the recommendation of acceptance. The camera-ready addresses the three points in the decision notice.
>
> > The reviewer 14oQ still noted that the overall scope remains somewhat limited.
>
> We added Section 7 (Discussion and Limitations). Paragraph 1 notes that our experiments span multiple baselines (Random+WFC and flat PPO), two physics solvers (FEM and FNO), and three scene scales (10×10, 15×15, 20×20). The values of target height, material, and loading scenario were chosen to make the task sufficiently challenging, as the low performance of the baselines confirms. The same paragraph also explains why the framework itself does not encode these parameters.
>
> > presentation clarity
>
> We added clarifications to the main text that previously appeared only in our responses to reviewers:
>
> - Section 6.3 now describes how each point in Figure 7 is computed on freshly sampled rollouts per seed, which addresses the held-out generalization question.
> - Paragraph 2 of Section 7 brings the shape-discovery vs. shape-imitation argument from our rebuttal into the manuscript and explains why Brick-by-Brick and BrECS are not used as empirical baselines.
>
> > additional discussion of limitations
>
> Paragraph 3 of Section 7 covers the remaining limitations (hand-designed reward, computational cost of physics simulation, simplified material and loading models, single brick type) and replaces the limitations paragraph that previously closed the Conclusion.
>
> We have also de-anonymized the manuscript, added a public code release at https://github.com/iSegments-Lab/stress_aware_bricks_model (linked from the abstract), and included an Acknowledgments section that discloses LLM use (Claude Sonnet for grammar, vocabulary, and clarity editing).